# Short-Term Cortical Electrical Stimulation during the Acute Stage of Traumatic Brain Injury Improves Functional Recovery

**DOI:** 10.3390/biomedicines10081965

**Published:** 2022-08-12

**Authors:** Liang-Chao Wang, Wei-Yen Wei, Pei-Chuan Ho

**Affiliations:** 1Institute of Clinical Medicine, College of Medicine, National Cheng Kung University, Tainan 701, Taiwan; 2Division of Neurosurgery, Department of Surgery, National Cheng Kung University Hospital, College of Medicine, National Cheng Kung University, Tainan 701, Taiwan

**Keywords:** traumatic brain injury, neural stem cells, cortical electrical stimulation, functional recovery

## Abstract

Functional restoration is an important issue in the treatment of traumatic brain injury (TBI). Various electrical stimulation devices and protocols have been applied in preclinical studies and have shown therapeutic potential for brain trauma. Short-term invasive cortical electrical stimulation during the acute stage of TBI might be a feasible adjuvant therapy for patients with moderate-to-severe brain injury receiving neurosurgical treatment in the intensive care unit. However, the therapeutic effects of short-term multisession cortical electrical stimulation for brain trauma are not clear. This study explored the therapeutic effects of acute-stage short-term cortical electrical stimulation on TBI. We conducted seven sessions of one-hour cortical electrical stimulation from day 0 to day 6 in rats after brain trauma by controlled cortical impact and then evaluated the functional outcome and histopathological changes. Our data showed that short-term cortical electrical stimulation improved motor coordination, short-term memory, and learning ability and attenuated neurological severity after brain trauma. Lesion volume, apoptosis, and gliosis after brain trauma were reduced, and trauma-induced neurogenesis in the hippocampus for the innate neural reparative response was increased. Our study demonstrated that short-term cortical electrical stimulation applied in the acute stage of traumatic brain injury is a potential adjuvant therapy to improve the recovery of neurological deficits.

## 1. Introduction

Traumatic brain injury (TBI) is one of the leading causes of death and disability in young and healthy individuals. Survivors of TBI often suffer from permanent functional deficits and thus impose a heavy socioeconomic burden on families and communities. Hippocampal-associated learning and memory impairments are common residual functional deficits following TBI and are among the most frequent complaints heard from patients and their relatives [1,2]. Despite decades of research, there are no clinically approved drugs for the treatment or prevention of cell death after TBI. The development of therapeutic strategies to attenuate functional deficits after TBI is an important issue.

The restoration of neurological function after brain injury involves neuroregeneration and neuroplasticity. Neural stem/progenitor cells (NSCs/NPCs) in the central nervous system (CNS) are self-renewing multipotent cells which can be activated and give rise to various types of neural cells under stimuli caused by CNS diseases or trauma, indicating innate efforts in neural repair [3,4]. However, the innate repair effort is insufficient for functional restoration. NSCs must proliferate, differentiate, and migrate to reorganize neural circuitry for functional recovery. Therapeutic approaches that provide optimal environments for neurogenesis and neuroplasticity are important for functional restoration after brain injury.

Alternative non-pharmacological procedures such as electrical and magnetic stimulation have been suggested as new therapeutic strategies to modulate neural activity and plasticity and restore deficits after brain injury. Several preclinical studies have demonstrated the potential of electrical stimulation in improving functional recovery after brain injury. Electrical stimulation can affect stem cell migration, proliferation, and differentiation by activating several intracellular signaling pathways and the intracellular microenvironment [5,6]. Electrical stimulation can promote the restoration of neurological function after CNS damage by modulating neuroregeneration and neuroplasticity [7]. In the acute stage after ischemic stroke, direct current stimulation has shown neuroprotective effects by attenuating ischemic injury [8,9]. In an animal model of TBI, electrical stimulation enhanced angiogenesis and the mobilization of endothelial progenitor cells to improve the recovery of cognitive deficits after TBI [10]. In combination with rehabilitation training, electrical stimulation enhances the improvement of motor function impairments and neuroplasticity after TBI [11]. Electrical stimulation may increase 5-hydroxytryptamine expression in the subventricular zone and attenuate the stroke-induced neurological deficits of rats [12]. The above evidence suggests that electrical stimulation may improve neural functional recovery through neuroprotection and neurogenesis-related processes, though the mechanisms involved in electrical stimulation-induced recovery are not well understood.

Various devices and stimulation protocols have been applied in preclinical studies to evaluate the therapeutic potential of electrical stimulation. Cortical electrical stimulation (CES) is an invasive stimulation method that can provide more precise and intense stimulation by indwelling electrodes to the target structure [13]. The stimulation protocol, such as the different positions of electrodes and neural targets, the current intensity, the duration, the timing of stimulation, the usage of the concomitant task, and the time point of the application of electrical stimulation, may influence the therapeutic outcomes of electrical stimulation. Various stimulation protocols have been tried in animal models and have shown their potential in the treatment of TBI [10,14,15]. These studies used 3 to 4 weeks of invasive CES. However, long-term invasive CES carries the risk of infection and is less feasible clinically. Park E. et al. reported that a single session of CES on the second day after TBI of rats can attenuate neuroinflammation and enhance neurogenesis [16]. In clinical scenarios, 1 to 2 weeks of a brain indwelling catheter for drainage or monitoring is a common neurosurgical therapeutic technique for the treatment of moderate-to-severe TBI. Short-term invasive CES in the acute stage of TBI might be a feasible adjuvant therapy for patients with moderate-to-severe brain injury receiving neurosurgical treatment in the intensive care unit. However, the therapeutic effects of short-term multisession CES are not clear. The impact of acute-stage short-term CES on the functional recovery of TBI deserves further study.

In this study, we applied short-term, multi-session CES during the acute stage of TBI and evaluated the impact of CES on functional restoration. We used the controlled cortical impact (CCI) model of TBI in rats, and seven sessions of one-hour CES were applied from day 0 to day 6 after TBI. Functional outcomes and histopathological changes were evaluated. We also explored the impact of this CES protocol on neurogenesis and related signaling pathways in the hippocampus. The results of this study will help to develop therapies using CES to promote functional recovery after TBI.

## 2. Materials and Methods

### 2.1. Experimental Protocol

All experiments used adult male Sprague–Dawley rats weighing 250–300 g, which were purchased from BioLASCO Taiwan Co., Ltd. (Taipei, Taiwan). The rats were kept in cages at an appropriate stocking density, with an adequate supply of feed and drinking water. The temperature, humidity, and light-dark cycle of the environment were automatically regulated. All experimental procedures were approved by the Institutional Animal Care and Use Committee (IACUC) (number 110137) of the National Chen Kung University and were implemented according to regulations. The rats were randomly allocated to three groups: sham operative (sham), CCI + sham stimulation (TBI), and CCI + CES (TBI + ES). The sham operative group received craniectomy prepared for CCI and the insertion of electrodes for CES, but neither CCI nor stimulation was conducted. This group was to validate the impact of operation procedures per se on rats’ healthy condition. The sham stimulation group received CCI and electrodes insertion but without stimulation; this group is the control condition for TBI with CES treatment. The experimental setup is illustrated in Figure 1.

### 2.2. Controlled Cortical Impact

CCI was conducted using the protocol suggested by the National Institutes of Health (NIH), with modifications [17]. We used a pneumatic-driven impactor device (RWD, model 68099) to produce the controlled mechanical deformation of the cortex, with the dura remaining intact. Surgery was performed under the surgical depth of anesthesia using a mixture of O_2_ and 1.5% isoflurane. The rat was placed on a stereotactic frame, with its head leveled and fixed by ear bars. A midline incision was made to expose the parietal bones. A 5 mm diameter skull bone window, with the center located 2.5 mm lateral to the midline and −3.5 mm posterior to the bregma, was carefully made with an electric drill bit to avoid dura mater injury. The rat was then transferred to the nose cone on the stereotactic frame of the impact device. A 3 mm impactor tip was fixed to the impactor. The X and Y coordinates of the head frame were adjusted to center the impactor tip over the hole. The impact conditions were set as follows: speed, 3 m/s; depth, 3 mm; dwell time, 0.3 s. After setting the impact condition, the distance to the dura was calibrated by lowering the impact tip until it was alarmed for minimal contact with the dura. Anesthesia was turned off for seconds before the impact to prevent apnea after the injury. After confirming the impact setting, the impactor was fired to slam down and then retract to produce controlled brain injury. Anesthesia was resumed after confirming the smooth breathing of the rats after CCI. After the CCI procedure, anesthesia was reversed, and the rats were returned to the housing room. In the sham TBI group, except for the firing of the impactor, the rats received all the procedures that the TBI group did.

### 2.3. Cortical Electrical Stimulation

The electrodes were made of 0.25 mm-diameter platinum- and iridium-annealed wire (platinum:iridium, 90:10 wt%) (Navantor, AF-39383-50CM) hooked to an electrical wire to connect it to the stimulator (Figure 1F). The cathode was inserted 3 mm below the cortical surface through a small hole at the 12 o’clock position of the craniotomy for cortical impact. The anode was placed at the 6 o’clock position. The electrode was fixed to the skull bone using two anchoring screws. The field of electrical current can reach 2~3 mm of the depth of the rats’ brains when the electric current is applied epidurally [18]; thus, the field of electric stimulation in this study can include the sensory-motor cortex and hippocampus. The electrical stimulation procedure was performed as in our previous study [8]. The electrodes were connected to a linear insulation stimulator (WPI, A395) with data acquisition hardware (PowerLab 8/35 with LabChart version 7.0, AD Instruments, Dunedin, New Zealand) that regulated the output of the stimulator. The electrical current settings were 20 Hz, 2 ms biphasic bipolar pulse, and 100 μA. This stimulation current was shown to have neuroprotective effects against ischemic brain injury in our previous study [8]. The rats received a 60 min session of cortical electrical stimulation under anesthesia every day from day 0 to day 6 post-TBI. For the sham stimulation, the electrodes were connected to the stimulator without power supply.

### 2.4. Neurological Function Test

We used an 18-point scoring system of the neurological severity score (mNSS) [19] to evaluate the neurological status of the rats on the third and seventh day after trauma. The content of the scoring systems is listed in Table 1. adapted from [19] 2001 Chen, J.

### 2.5. Rotarod Test

On the seventh day post-TBI, motor coordination was evaluated using the rotarod test [8,20]. The training started three days before the operation was performed, and three trials were performed per day. Each rat was placed on a rod (Panlab, LE8505, Harvard apparatus, Barcelona, Spain) that continued to rotate with accelerating speed from 4 rpm to 40 rpm for 5 min until the rat could not keep up with the speed of rotation and fell or the timing ended. In the formal test, the rats were tested each time for 5 min for a total of five trials, and the latency to fall was recorded. The average score of the five tests was used for the statistical analysis.

### 2.6. Morris Water Maze Test

The Morris water maze (MWM) learning task was used to evaluate learning abilities and memory after TBI [21]. A complete test contains a total of six sessions over six days and is performed by observers blinded to the experimental groups. The animals were subjected to four trials per session, one session per day, from the 9th to the 14th day after TBI. In each session, the animals were randomly placed at one of four different starting positions, which were equally spaced around the perimeter of the pool. The rats were allowed to find the platform within 120 s. If the testing animal failed to find the platform within 120 s, the session was terminated, and the rats were guided to the platform. The rats were allowed to stay on the platform for 20 s after the completion of a session. The time spent by individual rats in acquiring the platform was recorded as the escape latency.

### 2.7. Two-Object Novel Object Recognition Task

A novel object recognition (NOR) test can evaluate changes in rodents’ short-term recognition memory in disease models or after drug administration. This experiment contains three trials. On the 15th day after the surgery, the rats were placed in an open-field box and allowed to adapt for 30 min. On the 16th day after the surgery, two identical objects were fixed in the box, and the rats were allowed to explore for 3 min. On the 17th day after the surgery, one of the two old objects was replaced with a new object, and the rats were allowed to explore for 3 min. A camera was used to record each time when the rat explored. When the rat’s nose touched or pointed to an object within a distance of 2 cm, we judged it to be exploring. The data were analyzed as a discrimination index, which is defined as follows: the exploration time at the novel object—the exploration time at the familiar object/the exploration time at the novel object + the exploration time at the familiar object [22].

### 2.8. Brain Sections Preparation

On the 18th day after TBI, the rats were sacrificed for tissue examination. To prepare the tissue for immunostaining, the rats were deeply anesthetized with 5% isoflurane and transcardially perfused with phosphate-buffered saline (PBS). The brain tissues were post-fixed in 4% paraformaldehyde (PFA) overnight, dehydrated, and paraffinized. The brains were removed and post-fixed for 4 h in 4% PFA, processed, and embedded in paraffin. Coronal sections (15 μm-thick) were taken at every millimeter from 2.2–6.3 mm relative to the bregma.

### 2.9. Tissue Staining Methods

#### 2.9.1. Cresyl Violet Staining

For the cresyl violet staining procedure, the slides were placed in heated (50 °C) 0.1% cresyl violet acetate for 20 min, and excess stain was rinsed off in running tap water. Finally, the sections were mounted on coverslips with a mounting medium and dried overnight. The injured area was quantified using HistoQuest analysis. The lesion size was evaluated using the following formula: percentage of the lesion size = (contralateral cortex area − ipsilateral spare cortex area)/(contralateral cortex area) × 100% [23].

#### 2.9.2. Immunofluorescence Staining

The detailed protocol for immunofluorescence (IF) staining has been previously described [8]. During the staining stage, the slices were deparaffinized, rehydrated, immersed in 0.01 M citric acid (100 °C) for 10 to 20 min for antigen retrieval, and finally blocked in PBS containing 0.5% Triton X-100 (Sigma-Aldrich, Burlington, MA, USA) and 5% normal donkey serum (Millipore, Temecula, CA, USA). The slides were then incubated overnight with primary antibodies. The following antibodies were used: anti-NeuN (Millipore; MAB377), anti-cleaved-caspase3 (Cell Signaling Technology, MA, USA; 9661S), anti-GFAP (Millipore; MAB3402), anti-BrdU (Abcam, Cambridge, UK; ab6326), anti-β-catenin (Sigma; SAB4500543), and anti-doublecortin (DCX) (Cell Signaling Technology; 4604S; Millipore; AB2253). The slides were washed with PBS containing 0.1% Tween-20 to remove the primary antibody, replaced with DAPI and the corresponding secondary antibody, and incubated for 1 h. Finally, the secondary antibody was washed, and the slides were sealed with mounting glue (Dako, CA, USA).

### 2.10. Image Analysis and Cell Counting

The image analysis of the brain slices was performed as previously described [8]. The images were captured using a fluorescence (BX51, Olympus, Tokyo, Japan) or confocal (FV3000, Olympus, Tokyo, Japan) microscope. The number of positive cells and the total stained area were obtained using TissueQuest software version 7.0 (TissueGnostics, Vienna, Austria). Three to four views of each rat’s ipsilateral cerebral hemisphere were randomly counted under a 20× objective in coronal slices.

### 2.11. Western Blotting

Western blotting was performed following the protocol described by Tsai et al. [24]. The perilesional cortex was identified to be 1 mm in thickness surrounding the lesion cavity, and the hippocampus was dissected. The total protein in the brain samples was extracted using a radioimmunoprecipitation assay (RIPA) buffer containing protease and phosphatase inhibitors (Roche, Basel, Switzerland). The protein concentration in the supernatant was measured using a BCA protein assay kit (Pierce, Rockford, IL, USA). For western blotting, we used 10–12% SDS PAGE for electrophoresis to separate the extracts according to their molecular weights. The separated proteins were transferred to nitrocellulose (BioTrace^TM^) or polyvinylidene difluoride (PVDF) membranes (Millipore) and blocked in 5% BSA-containing PBS with 0.1% tween-20 (PBST) for 1 h. After blocking, the membranes were hybridized overnight with the primary antibody. The following antibodies were used: ERK (Millipore; 06-182), p-ERK (Millipore; 05-797), Akt (Cell Signaling Technology; 2920), p-Akt (Cell Signaling Technology; 9271), anti-glial fibrillary acidic protein (GFAP) (Millipore; MAB3402), β-catenin (Sigma; SAB4500543), and actin (Millipore; MAB1501). The membrane was washed with PBST to remove the primary antibody and then incubated with the secondary antibody for 1 h. The blots were visualized using Western Lighting Plus-ECL (PerkinElmer, Waltham, MA, USA). The quantification was performed using ImageJ software, and the results were normalized to β-actin and expressed as a percentage of sham.

### 2.12. Statistical Analysis

Statistical analysis and graphical representation were performed using GraphPad Prism 8 (GraphPad Software, San Diego, CA, USA) for all data and are expressed as the mean ± standard error of the mean (SEM). The escape latency in the Morris water maze task was analyzed using repeated measures analysis of variance (ANOVA) with a post hoc Tukey’s test for multiple comparisons. One-way ANOVA, followed by Tukey’s post hoc test, was used to compare other independent experiments with multiple groups. A two-tailed Student’s *t*-test was used to analyze the two-group comparisons, and the Mann–Whitney U test was used for non-normally distributed data.

## 3. Results

### 3.1. CES Improves Sensory-Motor and Cognition Functional Recovery

The neurological function of TBI rats was evaluated by mNSS, which comprises balance, sensory, motor, and reflex tests. Three days after the surgery, both TBI groups had significantly higher mNSS than the sham-operated group. The difference between the CES-treated and sham-stimulated groups was not significant, suggesting no significant intergroup differentiation in TBI severity (Figure 2A). On day 7 after TBI, all rats showed a partial recovery of neurological function, whereas the CES group had a significantly lower mNSS than the sham-stimulated group (Figure 2B). These data demonstrate that CES can improve neurological recovery.

At the seventh day post-TBI, the rotarod test was used to evaluate motor coordination in rats. The latency to fall in the CES-treated group was significantly longer than that in the sham-stimulated group (*p* < 0.001) (Figure 2C), indicating that CES improved motor coordination after TBI.

NOR tests were conducted from day 15 to day 17 post-TBI to evaluate short-term recognition memory. The discrimination index was used to evaluate the ability of rats to differentiate between familiar and new objects. Our data showed that the CES-treated group had a significantly higher discrimination index than the control group (*p* < 0.001) (Figure 2D), suggesting that CES can improve the recovery of short-term cognitive function after TBI.

We conducted the MWM tests from day 9 to day 14 after TBI and used the escape latency to evaluate the learning abilities of rats. The CES-treated group had a significantly shorter escape latency than the control group from day 2 to day 6 of the test. There was no significant difference between the sham-operated and CES groups (Figure 2E). The data showed that CES improved the recovery of learning ability after TBI.

Altogether, CES reduced neurological severity and improved motor coordination, short-term memory, and learning ability recovery after TBI (Figure 2). CES applied during the acute stage of TBI can improve functional recovery after TBI and is a potential strategy to attenuate neurological deficits after brain injury.

### 3.2. CES Reduces Lesion Volume and Attenuates Apoptosis in the Delayed Post-Traumatic Period

To evaluate neuroprotection using our CES protocol, we evaluated lesion volume after TBI with or without CES. The rats were sacrificed on day 18 after TBI, and the lesion volume was calculated using cresyl violet staining. Our data showed a significantly decreased lesion volume in the CES-treated group (*p* < 0.05) (Figure 3), suggesting neuroprotection by our CES protocol (Figure 3). Apoptotic cell death plays an important role in secondary injury after TBI, and caspase-3 is an important player in apoptosis. We evaluated the number of cells expressing cleaved caspase-3 (cCas-3) as an indicator of apoptotic cells. The number of apoptotic cells in the perilesional cortex was significantly decreased in the CES-treated group (*p* < 0.005) (Figure 4), indicating that CES during the acute stage of TBI can attenuate apoptosis during the delayed post-traumatic period. GFAP is a clinical indicator for severity and prognosis after TBI [25,26] and is also an indicator for gliosis after CNS injury. Gliosis after CNS injury may interfere with the reorganization of the neural circuitry and functional recovery. GFAP expression in the perilesional cortex was evaluated using immunohistochemistry staining and western blotting. Our data demonstrated that GFAP expression was significantly decreased in the CES-treated group (*p* < 0.05) (Figure 5). CES in the acute stage of TBI may attenuate apoptosis and gliosis in the delayed posttraumatic period, thus improving the microenvironment for functional recovery.

### 3.3. CES Promotes TBI-Induced Hippocampal Neurogenesis

To evaluate the impact of CES on hippocampal neurogenesis, we used BrdU to label newborn cells in the dentate gyrus of the hippocampus. Both TBI groups had significantly more BrdU-labeled cells in the hilus and granular cell layer (GCL) than the sham group, indicating TBI-induced cell proliferation. Moreover, the number of BrdU-immunoreactive cells in the CES-treated group was significantly higher than that in the control group, suggesting that CES can further improve TBI-induced cell proliferation (Figure 6B). Next, we used BrdU and NeuN costaining to evaluate the number of mature newborn neurons. Nevertheless, the increased number of co-stained cells indicated that TBI induced neurogenesis in the dentate gyrus, whereas CES further improved TBI-induced neurogenesis (Figure 6C). NSCs reside in the subgranular zone (SGZ) of the dentate gyrus of the hippocampus. Immature neurons derived from the SGZ migrate outward to the GCL and differentiate into mature neural cell phenotypes or fail to mature and die. To examine the influence of TBI and CES on immature neurons, we examined the number of immature neurons by identifying DCX-immunoreactive (DCX^+^) cells. Our data showed that the number of DCX^+^ cells in the GCL of the dentate gyrus did not differ significantly between the groups (Figure 7A,B). Since immature neurons migrate from the SGZ to the GCL for maturation, we further examined the number of DCX^+^ cells residing in the GCL. We found no significant difference in the number of DCX^+^ cells between the TBI and sham groups. However, in the CES-treated group, we found a significant increase in DCX^+^ cells in the junction zone of the GCL and SGZ compared with the TBI-only and sham groups (Figure 7C,D). These results indicate that CES can stimulate the proliferation and migration of immature neurons from the SGZ to the GCL. Altogether, our data indicate that CES stimulates neurogenesis in the dentate gyrus of the hippocampus and helps improve cognitive deficits after TBI.

### 3.4. CES Upregulates Neurogenesis Signal Pathways

The PI-3 kinase/Akt, MEK/ERK, and Wnt/β-catenin signaling pathways play important roles in regulating neurogenesis and differentiation. We examined the impact of TBI and CES on these signaling pathways in the hippocampus. The immunofluorescence staining of β-catenin and DCX in the dentate gyrus of the hippocampus showed increased β-catenin and DCX double-stained cells in the TBI + ES group (Figure 8A). We used western blotting to quantify the ratio of phospho-Akt to total Akt, the ratio of phospho-ERK to total ERK, and the amount of β-catenin and then compared them with those in the sham group. Our data demonstrated that the amounts of phospho-AKT and β-catenin in the hippocampus decreased on the 18th day after TBI (Figure 8C); however, phospho-ERK increased (Figure 8D). CES treatment upregulated phospho-AKT and β-catenin but did not change the phospho-ERK signal (Figure 8E). Our results suggest that CES might upregulate neurogenesis signaling pathways and promote an innate reparative response towards neural injury. However, the detailed mechanisms by which CES influences these signaling pathways were not explored in the present study.

## 4. Discussion

In the present study, we used a rat model of focal TBI to evaluate the implications of short-term multi-session CES applied during the acute stage of trauma for functional recovery after TBI. We applied a 60 min session of biphasic bipolar pulsed electrical stimulation from day 0 to day 6 after the induction of TBI and found that 7 days of intermittent CES in the acute stage after TBI improved mNSS, rotarod, MWM, and NOR performance, indicating improved sensorimotor and cognitive function recovery. Histological examination revealed that the lesion volume and expression of cCas3 were significantly decreased in the CES-treated group, suggesting neuroprotection by attenuating apoptosis. We also observed an increase in cell proliferation and newly generated mature neurons in the hippocampus. The number of immature neurons at the junction of the SGZ and GCL increased, indicating that CES stimulated the migration of immature neurons out of the SGZ for maturation. The PI-3 kinase/Akt, MEK/ERK, and Wnt/β-catenin signaling pathways were also upregulated, suggesting that CES stimulates neurogenesis for neural restoration. Our data support the idea that our CES protocol improves hippocampal neurogenesis and the restoration of cognitive function. Short-term invasive CES during the acute stage of TBI is a promising therapy for improving functional recovery after TBI. 

### 4.1. CES Applied in the Acute Stage Improved Functional Recovery after TBI

In the present study, we conducted biphasic bipolar pulsed electrical stimulation using electrodes implanted into the perilesional cortex. Several preclinical studies have demonstrated the therapeutic potential of CES for TBI. Four weeks of CES treatment in TBI rats improved locomotor function and spatial memory recovery [14,15]. Three-week electrical stimulation may enhance angiogenesis and improve the cognitive deficits induced by TBI [10]. Nine weeks of concurrent daily motor training and 100 Hz monopolar CES improved forelimb motor performance [27]. These studies used long-term CES with different stimulation parameters of frequency, polarity, and current intensity. In the present study, we used biphasic bipolar pulsed currents for CES. Biphasic electrical stimulation prevents charge accumulation, generates lower levels of electrolysis products at electrodes, and can be applied for extended periods and at higher voltages; thus, it is more favorable than monophasic stimulation [28]. In addition to the stimulation current parameters, the starting time and duration of electrical stimulation may influence the effect of CES. The optimal stimulation protocol for CES for the treatment of TBI is unclear. Our study used a daily 60 min stimulation from the day of TBI to the sixth day after trauma and demonstrated the therapeutic effects of short-term CES on TBI. One to two weeks of indwelling intracranial catheters for drainage or monitoring is a common neurosurgical treatment for acute TBI. Our invasive short-term multisession CES protocol is feasible in the acute stage of TBI and is a potential adjuvant therapy for improving functional recovery. 

### 4.2. CES Attenuates Apoptotic Cell Death in the Subacute Stage of TBI

Apoptosis plays an important role in secondary brain damage. Accumulated preclinical studies have shown that apoptotic neuronal cell death occurs soon after TBI. After CCI, apoptosis is most apparent in the injured region and the hippocampus between 24 and 48 h [29,30,31]. Neuronal degeneration after CCI may last for at least one year [32]. The upregulation of cCas3 in the white matter of the corpus callosum can be observed three months after CCI in rats [32]. After CCI, the colocalization of caspase-3 and tau accumulation indicates a possible correlation between apoptosis and neurodegeneration in the TBI phase [32]. These studies suggest that apoptosis in the subacute stage of TBI may influence functional recovery.

Our present study showed that short-term multi-session CES can decrease the volume of damaged cortical cavity examined on the 18th day after brain contusion, as well as decrease cCas3 expression. These data suggest that CES applied during the acute stage of TBI may attenuate apoptosis during the subacute stage. The number of GFAP-positive astrocytes in the contused cortex reaches the peak on the 10th day after CCI [33]. Gliosis surrounding the contusion cavity, rather than neurogenesis, may decrease the volume of the injured cavity but does not help in functional recovery. Our data also showed that GFAP expression surrounding the contusion cavity decreased significantly in the CES group, indicating that the decreased lesion volume was due to the neuroprotective effects of CES. Taken together, CES applied in the acute stage of TBI has neuroprotective effects and can attenuate the progression of brain damage. 

### 4.3. CES Augments the Innate Reparative Mechanism after TBI by Inducing Neurogenesis of the Adult Hippocampus

Previous studies have demonstrated that TBI promotes NSC proliferation through innate repair and/or plasticity mechanisms [34,35,36]. In the adult hippocampus, newborn neurons developed from the SGZ migrate radially from the SGZ into the GCL and integrate functionally into the existing neural network [37]. The survival, differentiation, and integration of newborn cells into the existing neuronal circuitry are important for functional restoration. 

However, the influence of TBI on hippocampal neurogenesis remains controversial. Several studies using different animal models of TBI have demonstrated that TBI alters hippocampal neurogenesis. In animal studies, TBI-induced neuronal stem cell proliferation is relatively transient and may differ according to the severity and mechanism of trauma. Hippocampal cell proliferation and neurogenesis increased after the CCI of rats and mice [38,39,40]. The variable severity of the injury may have different effects on neurogenesis. Gao et al. reported that mild TBI induces NSC proliferation, peaking at 44–48 h after TBI [39]. Wang et al. reported that mild CCI did not affect neurogenesis, moderate CCI promoted NSC proliferation without increasing neurogenesis, and severe CCI increased NSC proliferation, immature neurons, and newly generated mature neurons [41]. However, Gao et al. also reported that, following CCI, the majority of cells proliferated are glial cells rather than neurons, and the number of new neurons generated following TBI was not significantly higher compared to the sham control [40]. Despite these differences, the evidence supports the idea that TBI stimulates the proliferation of NSCs in mature rodent brains. The evidence also supports the idea that injury-induced adult-born granular cells can integrate into the existing hippocampal circuitry [35,42], and this endogenous neurogenesis is directly associated with innate cognitive recovery after TBI [43,44]. However, the injury-induced innate reparative response is insufficient to restore the population of damaged or destroyed neurons and results in functional deficits after TBI. Newborn cells in the dentate gyrus of the hippocampus of rodents die within one to three weeks after birth [45], mostly in the first four days of their birth [46]. Under normal conditions, less than 25% of newborn neurons survive, become mature neurons, and synaptically integrate. Providing friendly microenvironments for the survival and maturation of newborn neurons and targeting the migration of NSCs to the appropriate area for the reconstruction of neural circuitry are essential for functional restoration. 

Several in vitro studies have investigated the migration of NSCs in the presence of electric fields. Charged balanced biphasic electric stimulation enhances the cathodal distance of cell migration in the corpus callosum [47]. Under biphasic monopolar pulse stimulation, NSCs migrate at a speed of 1 μm to the cathode, whereas biphasic bipolar pulse electrical stimulation promotes both the proliferation and differentiation of fetal NSCs [48] but non-directed migration [28]. We did not explore whether CES can direct newborn neurons to the lesion site because we used bipolar stimulation to promote the proliferation and differentiation of NSCs rather than to guide their migration. In addition, our observation time was not long enough for the new mature neurons to integrate into the reserved cortex. 

In this study, we used CCI to evaluate hippocampal neurogenesis after TBI with or without CES. On 18th day after TBI, we did not observe an increase in the number of DCX^+^ immature neurons in the GCL, whereas the number of immature neurons in the junctional zone between the GCL and SGZ increased. DCX is transiently expressed in immature neurons and is associated with their migration. Once cells become older and begin expressing markers for mature neurons, DCX downregulates to undetectable levels [49]. Our observations support the idea that TBI stimulates the proliferation and migration of immature neurons to the GCL for maturation, and CES further improves this effect. However, there was no overall increase in the number of DCX^+^ cells in GCL. Immature neurons in the hippocampus are the most vulnerable to TBI. In the CCI model, TBI induces the significant death of immature neurons in the dentate gyrus early post-injury, before injury-enhanced cell proliferation is observed [50,51]. Moderate traumatic brain injury triggers the rapid necrotic death of immature neurons in the hippocampus within 24 h, and a lower level of cell death in the hippocampal dentate gyrus may last for as long as 2 weeks [52]. The TBI-induced production of immature neurons must replenish their loss after trauma. Moreover, immature neurons must integrate into pre-existing circuitry to survive and become mature neurons, or they will fail to mature and die. These reasons might have resulted in no net increase in the DCX^+^ cells in our study. Our data showed that on the 18th day after CCI, there was a significant increase in the dispersed distribution of newborn cells in the GCL and hilus of the dentate gyrus. The number of newly generated mature neurons, as indicated by the co-staining of BrdU and NeuN, was also increased, suggesting that TBI induces hippocampal neurogenesis, whereas CES treatment further induced an increase in newly generated neurons, indicating that our CES protocol can stimulate endogenous neurogenesis. These data support the idea that our CES protocol, applied during the acute stage of TBI, can strengthen the TBI-induced innate reparative response and improve functional recovery. However, whether newborn neurons were integrated into the functional circuitry was not explored in our experiments. 

### 4.4. CES Upregulates Signal Pathways Regulating Cell Proliferation and Differentiation in the Hippocampus

The PI-3 kinase/Akt, MEK/ERK, and Wnt/β-catenin signaling pathways play important roles in regulating cell survival, apoptosis, proliferation, and differentiation. There are time- and region-dependent alterations in phospho-Akt and ERK signals in the acute stage of traumatic brain injury [53,54]. Phospho-Akt is diminished on the first day post-injury [55]. In the CA1 region of the hippocampus, phospho-Akt levels increase after TBI. Akt was significantly decreased as early as one hour after trauma; however, phosphorylation was accelerated by the fourth hour. The ratios of phospho-Akt/Akt in the paraventricular nucleus were significantly reduced on post-injury day 7 [56]. These changes are related to cell death and anti-apoptosis [57]. The present study evaluated the change in phospho-Akt levels on the 18th day after TBI and found that phospho-Akt decreased, while phospho-ERK increased in the hippocampus. Apoptosis is most severe in the acute stage after TBI and gradually declines thereafter. PI-3 kinase/Akt and MEK/ERK are also involved in modulating neurogenesis. Therefore, the change in phospho-Akt and phospho-ERK signals in the delayed post-traumatic period may be related to the regulation of neurogenesis.

The ERK and Akt signaling pathways are important players in controlling the proliferation and differentiation of NSC, and the interaction between ERK-Akt signaling pathways helps to maintain the self-renewal/differentiation balance of stem cells [58]. Akt is directly responsible for driving fetal neuroprogenitor cell proliferation, whereas ERK signaling is involved in the neuronal differentiation of NPCs [59]. ERK plays an essential role in neurogenesis and the survival of differentiated neuronal cells and serves as a switch to shift the self-new/differentiation balance [60]. The Wnt/GSK3/β-catenin signaling pathway is also an important modulator for nervous system development [61,62]. There is a strong expression of Wnt3 in the hilar cells of the dentate gyrus and active GSK3/β-catenin-signaling in the adult SGZ and GCL. Astrocyte-derived Wnt signaling modulates the proliferation and neuronal differentiation of adult-derived NPC via the β-catenin pathway [63]. Mao et al. also reported that Wnt/β-catenin signaling enhances the proliferation rather than the differentiation of adult NSCs [64]. The Wnt/GSK3β/β-catenin pathway also contributes to electric field-induced cell migration through the coordination of GSK3β phosphorylation, β-catenin activation, CLASP2 expression, and asymmetric redistribution to the leading edge of the migrating cells [65]. The above evidence indicates that TBI-induced changes in the PI-3 kinase/Akt, MEK/ERK, and Wnt/β-catenin signaling pathways are related to the modulation of neurogenesis and migration. Our data demonstrated that the downregulation of phospho-Akt and β-catenin signals after TBI was restored by CES, but the upregulation of phospho-ERK signaling after TBI was not affected. These results suggest that CES upregulates neurogenesis-signaling pathways for neural restoration. However, detailed molecular mechanisms were not explored in the present study.

## 5. Conclusions

The present study demonstrated the therapeutic potential of short-term cortical electrical stimulation applied during the acute stage of TBI to improve functional recovery. Our data showed that short-term CES improved motor coordination, short-term memory, and learning ability and decreased neurological severity scores after TBI. Lesion volume, apoptosis, and gliosis in the delayed post-traumatic period were also attenuated. CES also promotes TBI-induced neurogenesis in the hippocampus during the innate neural reparative response. Although the detailed mechanisms have not been explored, our study demonstrated that short-term invasive CES helps in neurological recovery and is a potential adjuvant therapy for improving functional recovery after TBI.

## Figures and Tables

**Figure 1 biomedicines-10-01965-f001:**
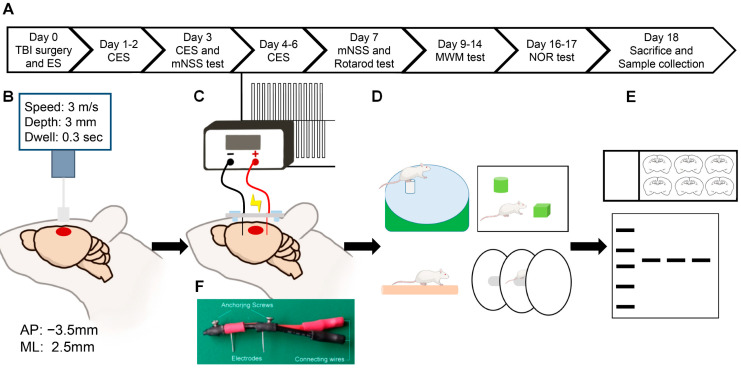
Schematic diagram of the experimental design: (**A**) Schedule of experiments, including traumatic brain injury (TBI) induction, cortical electrical stimulation (ES), modified neurological severity score (mNSS), rotarod test, Morris water maze (MWM) test, and novel object recognition (NOR) test. (**B**) Parameters of control cortical impaction and coordinates of craniotomy. (**C**) Parameters of CES. (**D**) Neurological test. (**E**) Histology and molecular analysis. (**F**) Demonstrating picture of the electrode device.

**Figure 2 biomedicines-10-01965-f002:**
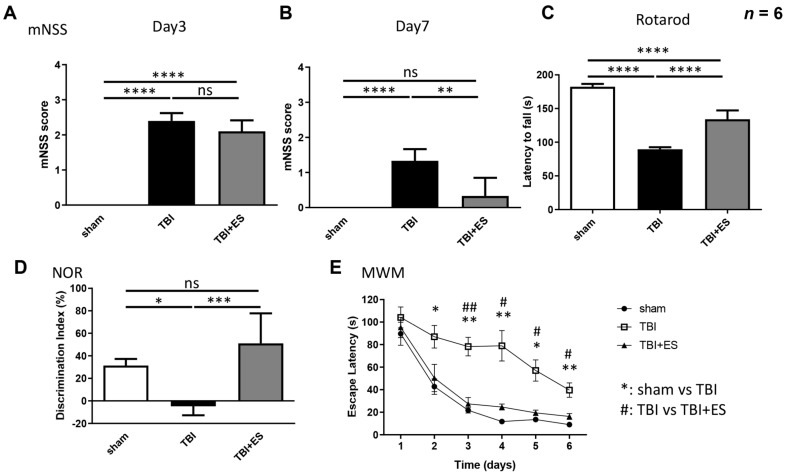
The neurological evaluation showed that CES improves functional recovery after TBI. (**A**) The modified neurological severity score (mNSS) evaluated on the third day after TBI showed no difference in terms of neurological severity between the TBI group (TBI + sham stimulation) and the TBI + ES group (TBI + cortical electrical stimulation). (**B**) On the seventh day after TBI, there was a significant improvement in mNSS in the TBI + ES group compared to the TBI group (*p* < 0.005). (**C**) The rotarod test showed a significantly prolonged latency to fall in the TBI + ES group (*p* < 0.0001). (**D**) The novel object recognition (NOR) test showed a significantly improved decimation index in the TBI + ES group (*p* < 0.001), indicating improved short-term memory by CES after TBI. (**E**) The Morris water maze (MWM) task showed that the escape latency in the sham group was significantly shorter than that in the TBI group from day 2 to day 5. The escape latency was significantly shorter in the TBI + ES group than in the TBI group, and there was no difference between the sham and the TBI + ES groups. Values are expressed as the mean ± SEM. **** *p* < 0.0001, *** *p* < 0.001, ** *p* < 0.005, * *p* < 0.05, ## *p* < 0.005, # *p* < 0.05.

**Figure 3 biomedicines-10-01965-f003:**
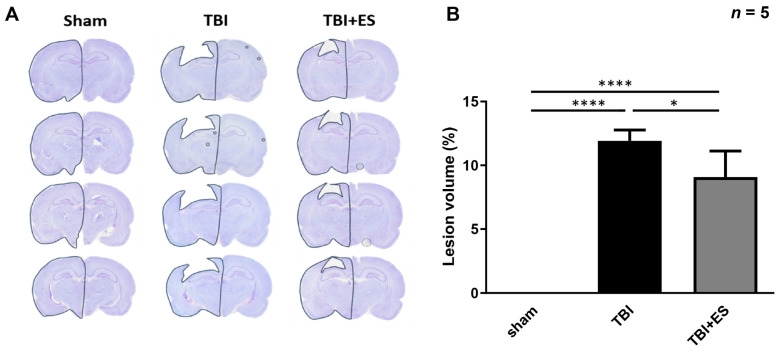
CES decreased the lesion volume of cortical contusion. (**A**) Representative pictures of cresyl violet staining. (**B**) Quantification of lesion volume. The data showed significantly decreased lesion volume in the TBI + ES group, suggesting neuroprotection by CES. Values are expressed as the mean ± SEM. **** *p* < 0.0001, * *p* < 0.05.

**Figure 4 biomedicines-10-01965-f004:**
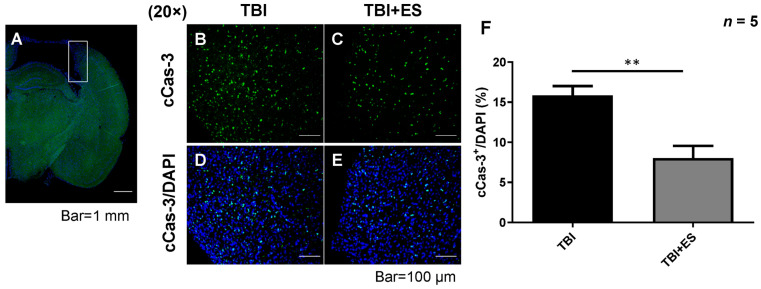
CES reduced cleaved caspase 3 (cCas-3) immunoreactive cells in the perilesional cortex. (**A**) Representative picture illustrating the sample region for the perilesional cortex. The cortical specimen was obtained within 1 mm from the wall of the lesion cavity. (**B**,**C**) Representative pictures of immunofluorescent for cCas-3 (green). (**D**,**E**) Representative pictures of merged immunofluorescent of DAPI (blue) with cCas-3. (**F**) Quantitative analysis showed a significantly decreased number of cCas-3 immunoreactive cells. Values are expressed as the mean ± SEM. ** *p* < 0.001.

**Figure 5 biomedicines-10-01965-f005:**
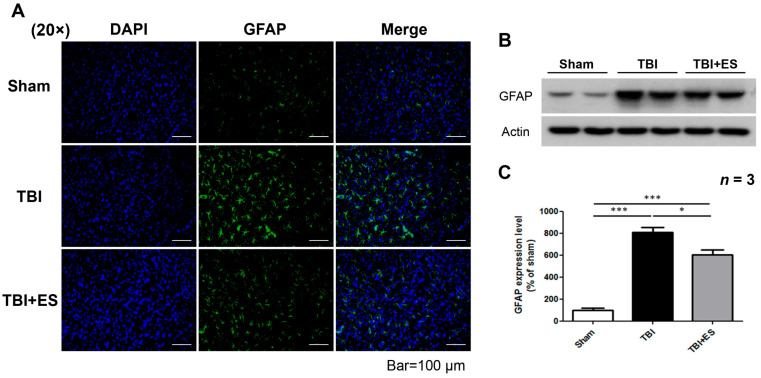
CES decreased the expression of Glial fibrillary acidic protein (GFAP) in the perilesional cortex. (**A**) Representative fluorescence images of GFAP (green), DAPI (blue), and merged picture. (**B**) Representative pictures of western blot for GFAP. (**C**) Quantitative analysis of western blot showed significantly decreased GFAP expression in the TBI + ES group compared to the TBI group. Values are expressed as the mean ± SEM. * *p* < 0.05, *** *p* < 0.001.

**Figure 6 biomedicines-10-01965-f006:**
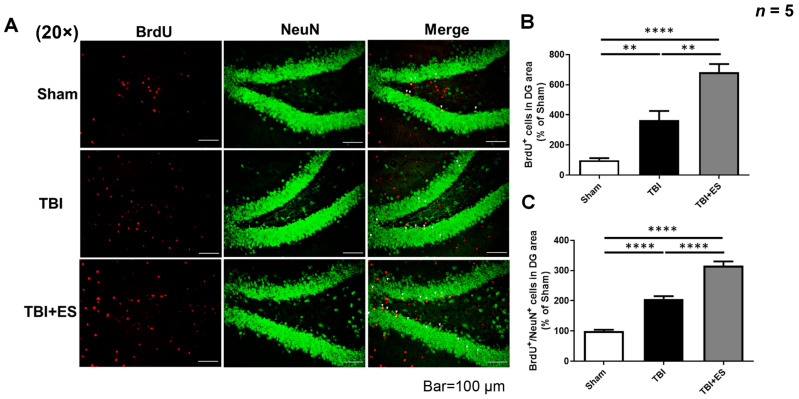
CES stimulates the proliferation of cells and increases the number of mature newborn neurons in the dentate gyrus of the hippocampus. (**A**) Representative fluorescence images of NeuN (green), BrdU (red), and merged picture. (**B**) Quantitative analysis showed significantly increased BrdU immunoreactive cells in the dentate gyrus. (**C**) Quantitative analysis demonstrated that BrdU/NeuN co-stained cells, indicating mature newborn neurons, are significantly increased in the TBI + ES group. Values are expressed as the mean ± SEM. **** *p* < 0.0001, ** *p* < 0.005.

**Figure 7 biomedicines-10-01965-f007:**
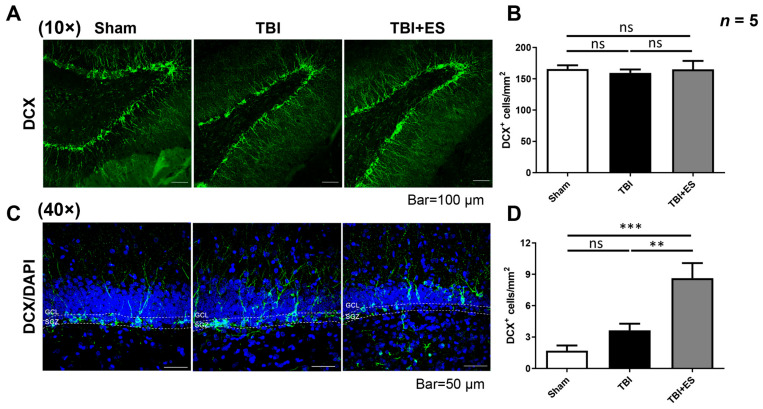
CES stimulates migration from the subgranular zone (SGZ) to the granular cell layer (GCL). (**A**) Representative pictures of the DCX immunofluorescence stain (green) of the dentate gyrus. (**B**) The quantification of DCX^+^ cells in the dentate gyrus showed no significant difference between groups. (**C**) Highly magnified fluorescence pictures of DCX (green) and DAPI (blue) in the junctional zone between SGZ and GCL. (**D**) Quantitative analysis showed no significant difference in the number of DCX^+^ cells in the junctional zone between the sham and TBI groups. In the TBI + ES group, there was a significant increase in DCX^+^ cells in the junctional zone between SGZ and GCL, indicating that CES stimulates the migration of immature neurons from SGZ to GCL. Values are expressed as the mean ± SEM. ns, no significance; *** *p* < 0.001; ** *p* < 0.005.

**Figure 8 biomedicines-10-01965-f008:**
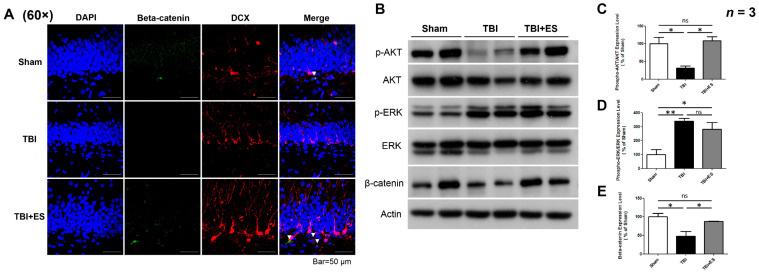
CES upregulates PI-3 kinase/Akt, MEK/ERK, and Wnt/β-catenin signaling pathways. (**A**) Representative fluorescence images of Beta-catenin (green), DCX (red), DAPI (blue), and merged picture. (**B**) Representative image of western blot. (**C**) Quantification of Akt phosphorylation presented as the ratio of phosphorylated Akt (phospho-Akt) to total Akt. The data showed significantly downregulated phospho-Akt on the 18th day post-TBI, whereas CES treatment restored the downregulation. (**D**) The quantification of Erk phosphorylation (phospho-Erk) demonstrated significantly upregulated ERK phosphorylation, but CES did not further increase the upregulation. (**E**) The quantification of β-catenin showed the significant downregulation of β-catenin expression after TBI and CES treatment returned the expression level. Values are expressed as the mean ± SEM. ns, no significance; ** *p* < 0.005; * *p* < 0.05.

**Table 1 biomedicines-10-01965-t001:** Components of the modified neurological severity score (mNSS) and scoring values.

Motor Tests		Score
Raising rat by tail		normal = 0; maximum = 3
	Flexion of forelimb	1
	Flexion of hindlimb	1
	Head moved >10° to vertical axis within 30 s	1
Placing rat on floor	Normal walk 0 Inability to walk straight 1 Circling toward paretic side 2 Falls down to paretic side 3	normal = 0; maximum = 3
Sensory tests		normal = 0; maximum = 2
	Placing test (visual and tactile test)	1
	Proprioceptive test (deep sensation, pushing paw against table edge to stimulate limb muscles)	1
Beam balance tests	Balances with steady posture 0 Grasps the side of the beam 1 Hugs the beam and one limb falls down from the beam 2 Hugs the beam and two limbs fall down from the beam, or spins on the beam (>60 s) 3 Attempts to balance on the beam but falls off (>40 s) 4 Attempts to balance on the beam but falls off (>20 s) 5 Falls off; no attempt to balance or hang on to the beam (<20 s) 6	normal = 0; maximum = 6
Reflex absence and abnormal movements		normal = 0; maximum = 4
	Pinna reflex	1
	Corneal reflex	1
	Startle reflex	1
	Seizures, myoclonus, myodystony	1

## Data Availability

The original contributions supporting the conclusions of this study are included in this article.

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
