# Peer review of "Short-Term Cortical Electrical Stimulation during the Acute Stage of Traumatic Brain Injury Improves Functional Recovery"

_biomedicines, 2022, doi:10.3390/biomedicines10081965_

Round 1

Reviewer 1 Report

This present study from Pei-Chuan Ho’s group investigated the effects of short-term cortical electrical stimulation (CES) in traumatic brain injury rat model. They applied CES protocol from day 0 to day 6 after TBI induction and found that CES can improve brain function, reduce lesion volume, cell death and astrocyte reactivity. CES increases cell proliferation, number of mature newborn neuron and stimulates cell migration in the dentate gyrus of hippocampus. The authors investigated possible protective molecular mechanisms, PI-3K/AKT, MEK/ERK and Wnt/β-catenin, that may be upregulated by CES. This study is well conducted and provides consistent findings. The manuscript is well written with appropriate references. There are minor concerns that need to be addressed to improve the quality of the manuscript.

1.      To increase the quality of the manuscript and to help readers keep track while reading, the authors should indicate and reference the figures throughout the manuscript. For example, “Three days after the surgery, both TBI groups had 290 significantly higher mNSS than the sham-operated group (Fig. 2A).

2.      In figure 8A and B, western blotting results show highly significant increased p-AKT and β-catenin levels in TBI+ES group compared to TBI group. With respect request, the author should add immunostaining results (similar to Fig. 6 and 7) to indicate the cell types that up-regulate those proteins in TBI+ES group. This will strengthen and expand their findings. Also, this will increase the number of the animal for this experiment where there were only 3 animals used for western blotting analysis.

Author Response

Thank you for your expert suggestions. We outline our brief point-to-point response of the revision. Please see the attachment.

Reviewer 2 Report

The manuscript reports on a study that used a rodent model of focal TBI to examine the effects of 60-minute sessions of cortical electrical stimulation over 7 consecutive days post injury on functional and histopathological outcomes. The study and findings are interesting, but revisions to the manuscript and more information is needed to fully evaluate its content. Specific comments are as follows:

1. The Abstract states that the “optimal stimulation protocol remains unclear.” This is not a parametric study nor does it have sufficient control to evaluate an “optimal stimulation protocol,” so the authors should be more specific as to what is unclear.

2. The Introduction, while informative, is lengthy and too broad. For example, it doesn’t appear that discussion of rTMS studies is warranted, and there is also some doubt about the need to address noninvasive electrical stimulation (e.g., tDCS) given the focus of the current study. Ultimately, the Introduction should be condensed to only include relevant information necessary for the reader to understand what is known and why this particular study was undertaken.  

3. Relatedly, the authors cite a few studies that have examined >3 weeks of CES and state that the effect of short-term CES has “seldom been investigated,” suggesting that it has been investigated. Why are the findings of those studies not discussed? The authors also seem to conflate short-term CES with CES administered in the acute stage. Again, the authors state that acute stage CES has “seldom been explored.” Please address studies that have examined short-term CES and those that have administered it in the acute stage. Much of the information in preceding paragraphs can be removed/condensed in favor of providing this information, which is highly relevant to the current study.  

4. A more explicit explanation of the three groups and the rationale for each in terms of the experimental objectives would be helpful early on (i.e., in section 2.1).

5. Why was the mNSS not administered at baseline? For the reader unfamiliar with this scoring system, it is unclear how much variability might be expected in the different functional domains (i.e., motor, sensory, reflex, and balance). Is it assumed that all rats across the various groups would have the same score? Why is no score presented in Figure 2 A and B for the sham group?

6. The stimulation parameters selected for the current study are justified as having shown neuroprotective effects against ischemic brain injury in the authors’ previous study. More justification is needed, especially since the goal is not the same in the current study; the goal is to evaluate therapeutic effects after TBI.

7. Which cortical region(s) was/were targeted by CES? There are several domains of function (e.g., motor coordination, memory/learning, etc.) and deeper structures (i.e., hippocampus) examined. Some rationale for how/why CES would be expected to have diffuse effects is warranted.

8. Was stimulation continuous over the 60-minute period? In other words, discrete pulse trains were not administered consecutively?

9. More justification is needed for the method of sham stimulation. There is really limited parametric information that can be gained from this study.

Author Response

(The authors gave the same response as above.)
